# Effects of Fouling Management and Net Coating Strategies on Reared Gilthead Sea Bream Juveniles

**DOI:** 10.3390/ani11030734

**Published:** 2021-03-08

**Authors:** Jordi Comas, David Parra, Joan Carles Balasch, Lluís Tort

**Affiliations:** 1Department of Cell, Biology, Physiology and Immunology, Universitat Autònoma de Barcelona, 08193 Bellaterra, Spain; David.Parra@hipra.com (D.P.); JoanCarles.Balasch@uab.cat (J.C.B.); Lluis.Tort@uab.cat (L.T.); 2Delta Aqua Redes, S.L.U. Morenot-Spain, 43870 Amposta, Spain; 3HIPRA, Av. De la Selva 135, 17170 Amer, Spain

**Keywords:** fouling, aquaculture nets, copper, cleaning strategy, *Sparus aurata*

## Abstract

**Simple Summary:**

Fish farming strives to cover the increasing demand for aquatic food sources as a result of global population growth. As a primary sector industry, aquaculture profit margins are narrow. Fouling management is an issue and represents a significant part of the operational cost of this activity. Over the last 30 years, this problem has been approached from different perspectives and the use of copper dioxide to control fouling production has been the most successful strategy. However, far beyond being the definite solution, the use of copper involves several concerns, and so the aquaculture industry has been continuously trying to find a reliable alternative. Coating the nets and cleaning on-site was adopted by the industry as a realistic alternative around 2015. This work contrasts these two fouling management strategies, simulating real working conditions by analysing the results from different perspectives. The conclusions from this work suggest a combination of both as a promising future alternative.

**Abstract:**

In aquaculture, biofouling management is a difficult and expensive issue. Cuprous oxide has been commonly used to prevent fouling formation. To cheapen net management and reduce the use of copper, the industry has proposed several alternatives. Currently, polyurethane coatings are being explored and commercially implemented. With this alternative, net cleaning is done in situ, reducing the number of nets necessary to raise a batch, thus ideally reducing operational costs. This pilot study compared this new strategy to the use of cuprous oxide. The results show that nets treated with antifouling perform better and bioaccumulation of copper in fish tissues do not pose health risks to fish. Alternatives involving on-site cleaning need to improve efficiency. Although the conditions of this work are not completely comparable to commercial aquaculture conditions, the results might indicate the strengths and constrains of the solutions tested in real life.

## 1. Introduction

Aquaculture is one of the fastest-growing industries globally. Its objective is to meet the ever-increasing demand for aquatic food sources, while also understanding fisheries’ progress and limitations. Cage aquaculture represents more than 65% of finfish production in Europe and the Americas with approximately 4.3 million tons produced in 2014 and is increasingly being introduced all over the world to places where conditions allow it [1].

The term fouling could be defined as the attachment and growth of living organisms such as bacteria, algae, bryozoans, and mollusks, among others, on surfaces immersed in the aquatic environment. Fouling formation is a natural phenomenon that occurs globally and is implicit in any activity developed at sea, including boats and aquaculture cages. Fouling growth is positively correlated with environmental factors such as water temperature, nutrient richness, and light hours. Its control has been pursued for a while [2]. In cage aquaculture, fouling is a major problem worldwide [3], difficult to manage, and expensive to control, representing an important part of fish farming production costs. Thus, fouling causes several maintenance and operational difficulties that affect not only the integrity of nets and other farm structures, but also fish performance and the health status of the site [4]. The main impacts of net fouling in cage aquaculture are: (a) Net occlusion, which reduces water exchange across the net, increases drag, and reduces dissolved oxygen levels and waste dispersal rate. These consequences become more severe as the size of the cages increase and so the ratio between net surface and net volume reduces. This increase in cage size is precisely a tendency nowadays in the salmon and in the Mediterranean industry. (b) Weight addition, which reduces buoyancy and cage volume, provokes fatigue, material stress, and eventually system failure [5]. (c) Acts as a reservoir for parasites and possible pathogens, increasing the incidence of potential disease and reducing fish performance [4,6,7], thus finally increasing costs and reducing the profitability of aquaculture companies.

Over time, several strategies have been used to control fouling formation in aquaculture nets. However, for the last two decades, after the ban of tributyltin (TBT) [8], antifouling paints containing cuprous oxide have proved to be highly effective against fouling formation and are widely used by the industry. These paints work by creating a toxic boundary layer at the surface of the paint as the component biocides leach, delaying and slowing down fouling onset [9]. Although eventually nets have to be replaced as they become occluded by fouling, antifouling treatments clearly prolong the lifespan of the nets.

Copper is a natural occurring element and essential in all eukaryote cells since it is a co-factor for many enzymes. However, it might also be toxic at high concentrations due to its redox nature [10]. Marine animals can uptake copper directly from their diet and by drinking water, and also across the skin and gills [11]. This could eventually lead to its accumulation in tissues and organs (such as gills and liver, among others) and put animals at risk, thus affecting consumer health and environmental balance. Thus, copper-based antifouling protects the net but may result in elevated levels of dissolved copper [12]. Although there is no indication that copper-based products will be prohibited in the near future [13], the industry is looking for more sustainable and cheaper solutions to reduce operational costs and impact to the environment. Several alternatives have been approached, among which on-site cleaning is one of the most developed. Although it is currently well implemented in the salmon industry and has been introduced in Mediterranean aquaculture, its effectiveness is now being argued, as was already pointed out when first being tested in the 1990s [14]. The main concerns are the cleaning rate achieved during the cleaning operation, netting abrasion produced by high water pressure, and the effects on fish health since, as in net changing operations, it can produce stress and additionally gill damage produced by the cloud of debris when fouling is washed off the nets.

On-site cleaning implies changes in net construction and management. The most obvious is that nets are cleaned in situ instead of being replaced and cleaned on land. The cleaning operation is done using water pressure cleaner disks (250–300 bar) and ideally only one net is used to raise a batch of fish to commercial size. In these nets, panels need to be more rigid than normal to ensure certain cleaning effectiveness. For that purpose, they are constructed with less slack than the other net panels, usually 2% to 3%. In order to protect twine from abrasion during the cleaning operations, nets are often coated with different materials. Currently the use of polyurethane coatings is being explored and yet implemented on an industrial scale. They do not prevent fouling formation but allow easy cleaning of the net. According to the literature, washing of nets is generally conducted every eight weeks in winter to two weeks in summer, but can occur as often as weekly during periods of high biofouling pressure [3]. Both net replacement and net cleaning operations might cause stress or damage to animals [7].

In view of these antecedents and due to a lack of field works in this area, the aim of this study was to compare in simulated real rearing conditions the performance of fish reared under two management strategies: (a) traditionally antifouling treated nets (involving the use of copper), and (b) polyurethane coated nets (cleaned on-site). To assess the performance of fish, the following indicators were assessed: growth, somatic indexes, copper accumulation in tissues, gill parasites occurrence, skin microbiota dynamics, and gill alterations. Additionally, the effect of on-site cleaning on the tensile strength of the nets was also analyzed. The aim of this work is to provide reliable information and contribute to better decision-making by the industry in fouling control strategy.

## 2. Materials and Methods

### 2.1. Study Area

The trial was carried out in a sea water pond located in the east coast of Spain, in the Ebre river delta (40°37′38” N; 0°39′41” E). It lasted 7 months, from December 2015 to June 2016. There was water circulation between the pond and open sea. The experiment simulated real rearing conditions in seawater cages.

#### Experimental Setup and System Description

Three experimental conditions (C1, C2 and C3) with three replicates each were set. Each replicate was allocated to a square 2.25 m^2^ polyethylene floating cage supporting a 210/96, 18-mm half mesh (The International Organization for Standadization ISO 1107, 2003), updated to (ISO 1107:2013) knotless nylon^®^ (PA6 light protected, Nexis, Slovakia) net 1.3 m deep, which provided a rearing volume of 2.92 m^3^. The system was laid in a way that the replicates in each condition followed the water flow across the pond. In C1, nets were treated with the commercial copper based antifouling paint Netwax NI3 (NetKem AS, Kolbotn, Norway) containing 11% in volume of cuprous oxide, (Cu_2_O). In C2, the nets were coated with a polyurethane layer (vi-cote™, Vicrapore, Scotland) containing 3% in volume of Cu_2_O, and in C3, the nets were coated with the same polyurethane layer without Cu_2_O. Antifouling treatment in C1 was applied through a vacuum system, allowing the paint to impregnate the netting. Prior to use, the antifouling paint was well homogenized using a paint mixer. In C2 and C3, the nets were dipped for 2 h in liquid polyurethane diluted solution (2:3 dilution). In C2, copper was incorporated in the polyurethane solution when treating the net. These procedures are usually followed by the net manufacturing industry. In all three conditions, the nets were air dried for 48 h after treatment. At the end of the experiment (7 months), the nets in C1 cages were changed to a new one in an operation that took about 2 h following the same methodology used in commercial cages, while C2 and C3 nets were cleaned on-site. During the experiment, C2 and C3 nets were also cleaned on-site every 40 days approximately since the beginning of the experiment. Every cleaning operation was performed at 250 bar using a diesel water pressure washer with a modified ending simulating the cleaner disks used in fish farming. This operation lasted about 20 min per cage, depending on the amount of fouling present in the net panels (Figure 1).

### 2.2. Fish Husbandry and Feeding

A total of 810 gilthead sea bream (*Sparus aurata* L. 1758) juveniles 75.50 ± 7.31 g provided from a local hatchery were acclimatized to the pond for fifteen days prior to initiating the experiment. The fish were then stocked into each cage at a rate of 90 fish per cage and maintained under natural conditions. They were manually fed twice a day ad libitum 5 days a week with a commercial diet of 46% crude protein, 19% fat (Skretting, Norway). From this batch of fish, 252 were used for analytical purposes.

#### 2.2.1. Water Quality Monitoring and Copper Determination in Water

Temperature and dissolved oxygen were daily measured at dawn and afternoon in the pond using a WTW Oxi 3205 portable oximeter calibrated to measure marine water. The same measurements were also taken weekly inside each cage. Nitrites, ammonia, pH, and salinity were measured weekly in the pond. Water chemical parameters were analyzed using a Hatch (USA) colorimeter. Salinity and pH were measured in an Inolab_IDS Multi 9310 station.

#### 2.2.2. Net Sampling and Image Analysis

Netting breaking strength (BS) was measured at the beginning and at the end of the experiment according to Nørsk Standard 9415 and ISO 1806 using a DYNA 300D/P dynamometer, Buraschi (Biassono, Italy) to compare the effect of washing the nets between the two fouling control strategies tested in the experiment.

To assess the effects of the experimental treatments (antifouling and coating) on the netting in rearing conditions, a total of 180 BS tests were performed at the beginning of the experiment on different pieces of experimental wet untreated net. A portion each of experimentally treated and untreated netting were placed into the experimental pond. At the end of the experiment, 45 BS tests were performed on each one of them.

To compare the effect of the different cleaning strategies on netting tensile strength, at the end of the experiment, 10 BS tests were performed on each net replicate (*n* = 30). The results were contrasted with those obtained in the BS tests performed at the end of the experimental period in the treated netting portions laid in the pond but not subjected to any cleaning process.

Fouling formation on the nets and the effect of on-site cleaning were measured according to previous works [5]. The percentage net aperture (PNA) and the percentage Net Occlusion (PNO) were calculated. PNA is defined as the area of netting not covered by twine or fouling and calculated as the ratio of the number of pixels, which represented mesh holes, to the total number of pixels of the image. Percentage Net Occlusion (PNO) at a given moment was calculated by using the equation suggested by Braithwaite et al. (2007).

To analyze PNA and PNO, pictures of eight panels per experimental condition and pictures of eight panels of experimental untreated netting were taken at the beginning of the experiment. Along the experiment, pictures of two netting panels of each cage (the ones north and south oriented) (*n* = 6) were also taken before and after every cleaning operation in C2 and C3. At the end of the experiment, pictures of two netting panels of each cage (the ones north and south oriented) (*n* = 6) were also taken from C1 cages just before net changing. The net panels’ images were captured by carefully lifting the nets. A white background was used to increase contrast. Pictures were always taken at the same distance of 28 cm. Images were digitally analyzed with the open source ImageJ 1.50i (https://imageJ.nih.gov/ij/ (accessed on 20 April 2017)) [15] and image processing was performed by thresholding in HSV (Hue, Saturation, and Value) format.

#### 2.2.3. Growth Assessment and Somatic Indexes

To quantify growth along the experiment, all the animals were weighed when setting the replicates of each condition and all surviving fish were weighed at the end of the experiment. Specific Growth Rate (SGR day-1) was calculated using the formula reported by Hopkins [16], where Wχ refers to the weight at the end of the experiment in g, W0 is the weight at the beginning of the experiment in g, and T is the time in days.

Growth was also compared to that obtained in a commercial fish farm under similar conditions. To do so, data from several actual sea bream batches reared and harvested in commercial cages in that Mediterranean area under the same temperature and feeding profile were used. Hepatosomatic (HSI) and spleen somatic (SSI) indexes were calculated at the beginning and at the end of the experiment, as previously described. Initial somatic indexes were calculated for 27 animals, (3 animals per cage, 9 animals per condition), and at the end of the experiment, somatic indexes were calculated for all surviving animals. Fish were sacrificed with 200 mg·L^−l^ MS222 (Sandoz, Germany), dissected, and their liver and spleen harvested and weighed. After that, those organs were immediately kept frozen at −80 °C for posterior use.

#### 2.2.4. Skin Mucus Collection and Preparation for Skin Microbiota Analyses

To assess skin microbiota changes during the experiment, skin mucus samples from 18 animals at the beginning of the trial and 18 more animals at the end of the experiment were taken. Fish were sacrificed with 200 mg·L^−l^ MS222 and skin mucus was quickly collected by gently scraping the dorso-lateral skin surface, as described in [17]. To separate skin bacteria from mucus, the cell-free supernatant was thereafter centrifuged at 10,000 rpm for 10 min at 4 °C. The resulting supernatant (containing skin mucus) was harvested and stored at −80 °C in Eppendorf tubes, whereas the pellet (containing skin bacteria) was re-suspended with PBS (pH 7.2) and centrifuged again at 10,000 rpm for 10 min at 4 °C. The resulting pellet containing the microbiota was collected and stored at −80 °C until the moment of the analysis. To identify the bacterial component of skin microbiota, an Illumina^®^ 16 s rRNA sequencing was performed.

#### 2.2.5. Gill Parasites Quantification

To quantify the parasites in the gills, twenty-four animals, were sacrificed at the end of the experiment with 200 mg·L^−l^ MS222 and individually frozen at −20 °C. For parasite quantification, once the frozen fish thawed, full gill arches were carefully excised and examined under stereomicroscope to determine parasites presence.

#### 2.2.6. Gill Histology and Gill Score Protocol

Histological observation was done on the gills at the beginning and at the end of the experiment (before, just after and 24 h after cleaning operations) to assess the effect of on-site cleaning on gill integrity. Seven animals at the beginning of the experiment and 15 at the end of the experiment were sacrificed using 200 mg·L^−l^ MS222; their complete gill arches were removed and fixed in 10% formalin for 24 h and finally stored in 70% ethanol. Afterwards, the gills were cleared in xylene in a tissue processor Leica TP1020 (Leica Biosystems, Nußloch, Germany) and paraffin-embedded in a paraffin embedding station Leica EG1150H following standard procedures. A microtome LEICA RM 2255 was used to obtain 5 µm sections, which were afterwards stained in Hematoxylin-Eosin. The microscopy examination was done with a NIKON Eclipse 80i (Nikon, Tokyo, Japan) microscope at 200× and 400× magnification.

Semi quantitative analyses was done on gill damage according to a protocol based on one developed by Mitchell [18], but slightly modifying some parameters. The score for the index parameters was based on the presence and extent of the following four primary parameters: lamellar hyperplasia, lamellar fusion, cellular anomalies (including degeneration, necrosis and sloughing), and lamellar edema with a score ranging from 0 to 3 (none, mild, moderate or severe) being assigned to each parameter. In addition, the presence (1) or absence (0) of these additional parameters were taken into account: cellular hypertrophy, inflammation, excessive numbers of MAST cells, circulatory disturbances (hemorrhage, telangiectasia, congestion), and excessive number of mucus cells. Thus, gill damage was measured by the score obtained: No (0–3), minor (4–6), moderate (7–9), and severe (>10) damage.

A total of 10 fields of gill sections from different animals of the three conditions were analyzed at the beginning of the experiment to obtain initial score. At the end of the experiment, 10 fields of each gill section from animals in C2 and C3 just before and after net cleaning were also analyzed. Finally, the gill status 24 h after net maneuver was conducted on 10 fields of gill samples from C2 and C3 animals.

#### 2.2.7. Copper Analysis

In order to assess copper transfer from experimental paints to water, five 50 L tanks containing: tank 1: pond water, tank 2: pond water + untreated netting, tank 3: pond water + C1 netting, tank 4: pond water + C2 netting and tank 5: pond water + C3 netting were kept and agitated regularly for the same period of time that the experiment lasted. Afterwards, copper content in the water was determined. Water samples were acidified with HNO3 (Merck Suprapur, Darmstadt, Germany) to reach a final sample medium HNO_3_ 1% (*v/v*). Copper determination was performed with inductively coupled plasma optical emission spectrometry (ICP-OES) in a PerkinElmer (USA)–Optima 4300DV spectrometer. For the analysis on tissues, a pool of samples from nine animals at the beginning of the trial (to obtain initial values) and twenty-seven animals, (3 animals per cage, 9 animals per condition) at the end of the experiment were taken. Fish were sacrificed with 200 mg·L^−1^ MS222, dissected, and the liver, gills, and a portion of dorsolateral muscle were removed and stored at −80 °C. For the analyses, 0.2–0.6 g of each sample was digested in a microwave oven (Milestone, Ultrawave, Italy) in concentrated HNO_3_ (Merck, Germany). Copper determination was performed using an ICP-MS spectrometer (Agilent 7500ce, Agilent Technologies, Santa Clara, CA, USA).

#### 2.2.8. Statistical Analysis

Statistical analysis was performed using the statistical package MINITAB (version 17), (Minitab, Inc., State College, PA, USA). All analyzed data were normally distributed. Normality was checked with the Anderson-Darling test. Growth and net features (net aperture, net occlusion and netting breaking strength) were compared using one way ANOVA considering “net treatment + net operation” as a single factor effect since one always depends on the other. Tukey’s tests were used to see specific differences between treatments considering significant differences at values *P* < 0.05. Copper accumulation in tissues, skin microbiota richness, and gill disorders were analyzed using the General Linear Model using time and treatment as factors to see differences between the beginning and the end of the experiment among groups. Whether differences were significant repeated, one way ANOVA and Tukey’s test were used for a particular factor. Statistically significant differences were assumed when *P* values were < 0.05.

## 3. Results

### 3.1. Water Quality

Water quality parameters during the experiment were well within a suitable range for rearing sea bream (Table 1 and Figure 2.). During winter months, temperature was below the optimal range for the culture of this species, but it did not pose a risk to the animals’ health. Sea bream culture in ponds in the area often experiences the same low temperatures profile during winter. A peak in nitrites and a low peak in salinity were detected in the pond, affecting all cages for a very short period of time.

### 3.2. Netting and Fouling Formation

Netting treatments and fouling control strategies tested in this work had different effects and displayed different results on netting tensile strength and fouling presence in net panels. Figure 3A shows Percentage Net Aperture (PNA) and Percentage Net Occlusion (PNO) during the experiment, as well as the effectiveness of on-site cleaning in C2 and C3. In all three conditions, netting treatment with experimental paints increased PNA, as already reported [7] due to twine contraction.

In conditions 2 and 3, the nets were cleaned in situ approximately every 40 days. After seven months, at the end of the experiment, and just before net operation (net changing or on-site cleaning), PNA was significantly lower in C2 and C3, 50.1% and 38.5%, respectively compared to C1, 62.9%. Subsequently, PNO was significantly higher in those conditions. Pre-cleaning PNA values in C2 and C3 recovered after cleaning up to 64% and 61% lower levels, respectively, although they were not significantly different from PNA at the beginning of the experiment. Within each condition corresponding to the on-site strategy (C2 and C3), the same level of mesh occlusion was recorded just before cleaning in all intermediate cleaning operations. The results clearly show the antifouling effect of the cuprous oxide incorporated in the C2 coating.

Wetting untreated netting resulted in a loss of breaking strength (BS). This loss was much higher after the period of seven months in sea water (Figure 3B). Antifouling paint and polyurethane coating seemed to protect the net since after seven months in sea water, breaking strength in all three conditions was higher than breaking strength in the control. According to the results obtained, cleaning operations deplete nets BS, regardless of the strategy used, but the effect of on-site cleaning at 250 bar caused much more damage to the net. While in C1, BS at the end of the experiment dropped from 102 kg to 92 kgs, in this experiment, after a rearing period of seven months and six episodes of on-site cleaning, breaking strength of those nets in C2 and C3 was significantly lower with values around 78 kg.

### 3.3. Growth and Somatic Indexes

Improving animal growth is the final goal of the aquaculture industry. In all cases, growth between February and May was below what should be expected according to the temperature profile during the experiment (Figure 4A). At the end of the experiment, growth achieved in C1 was significantly higher than the one achieved in the other two experimental conditions. Over the experiment, hepatosomatic index (HSI) decreased in fish of all experimental conditions. However, at the end of the experiment, HSI was significantly higher in C2 and C3 than in C1. Spleen somatic index (SSI) on completion of the experiment was significantly higher in C2 and C3, compared to C1 and to the initial value seven months prior (Figure 4B).

### 3.4. Gill Parasites and Skin Microbiota

To determine whether the presence of fouling impacted the parasite occurrence within the fish population, the number of gill parasites at the end of the trial was analyzed. Following the same hypothesis, the richness of fish skin microbiota was also assessed throughout the experiment. No evidence of gill parasites was found in C1 at the end of the experiment, whereas some adult parasites were found both in C2 and C3 (Table 2). Looking at the evolution of skin microbiota richness under conditions C2 and C3, in which substantial fouling formation was recorded along the experiment, an increase in richness of skin microbiota expressed as an increase of Shannon’s index, was found in the skin mucus. Significant differences were also found within the treatments along time and between conditions at the end of the experiment (Figure 4C).

### 3.5. Gill Histology

The results of this experiment showed very clearly the effect of on-site cleaning on gill integrity. In C1, significant differences were found just after net changing at the end of the experiment comparing with gill health status at the beginning of the experiment. The initial gill score of 2.78 (no damage) increased to 4.89 (minor damage). A general slight level of tissue disruption was detected, especially in very apical areas, but irrelevant lesions were detected. On the other hand, also at the end of the trial, in C2 and C3, highly relevant and significant differences were found between the initial and final status of gill integrity. Gill damage after net cleaning scored 11.6 and 9.20 in C2 and C3, respectively. In both cases, gills showed severe evidence of gill disorder with high levels of damage and deterioration, including primary and secondary lamellae hyperplasia, edema, and fusion and circulatory disturbances. Gill integrity status in C2 and C3 just before the last cleaning operation suggests that after the time-lapse period between cleaning episodes (40 days approximately), the gills showed a slight degree of recovery. Our results did not show any significant symptom of recovery 24 h after the cleaning operation (Figure 4D).

### 3.6. Copper Analysis

At the end of the experimental period, the results suggest that there was some copper release from the nets, which eventually accumulated in tissues. The test performed in order to assess copper release from experimental paints to the water (see Materials and Methods), showed that it did occur. While no copper was detected in the tanks containing pond water or pond water containing untreated netting, a concentration of 268 mg·L^−1^ was registered in the tank containing pond water + C1 netting, whereas a much lower concentration was recorded in tanks containing water + C2 netting and water + C3 netting, 15 5 mg·L^−1^ and 13.9 mg·L^−1^ respectively.

The amount of copper detected in tissues was higher in the liver than in the gills. Significant differences were found when comparing copper concentrations between the beginning and the end of the experiment. These differences where more evident in the liver than in gill (Figure 4E). In the liver, copper content at the end of the experiment was significantly higher than the initial values obtained from a pool of samples of nine animals (three animals per condition). Looking into the gills, only those corresponding to fishes in C1 displayed significantly higher concentration of copper at the end of the experiment compared to the initial values. In C2 and C3, there was a slight increase of the metal present in gills but without significant differences. There were no differences between treatments at the end of the experiment. Although not represented, copper content in muscle was lower than 0.25 µg·g^−1^, below the technique detection threshold.

## 4. Discussion

In aquaculture research, experimental work is often conducted under laboratory conditions and results are sometimes difficult to compare with what really happens in the field. In this work, our aim was to set up an experiment to see the effect of two fouling control strategies under real working conditions, so the obtained results could reflect what happens in commercial fish farms. Fish husbandry in open sea is often difficult since experimental facilities are not always available. In this experiment, field work was carried out in a sea water pond next to the sea to ensure optimal fish husbandry, to allow fouling production under natural conditions, and to facilitate sampling.

Our results show that antifouling coating in C1 avoid fouling formation, while polyurethane coatings in C2 and C3 did not prevent fouling formation, although, after image analysis, the effect of the small amount of antifouling present in C2 treatment was clearly detectable, resulting in a much lower PNO in C2 than in C3. Thus, as already stated for other copper-based paints [4,5,6,7], antifouling treatment in the present experiment, at both concentrations used, were highly effective. Considering the amount of resources needed at the industrial scale and the necessary cost-effectiveness of the operation, cleaning in C2 and C3 was performed approximately every 45 days from December to April when temperatures were below 18 °C and approximately every 30 days from May to the end of the experiment, with higher temperatures, longer days, and more fouling production. Net panels in experimental cages were constructed according to industry procedures to avoid pockets due to excessive slack to ensure good cleaning efficiency when cleaning on-site. However, complete cleaning in C2 and C3 was not achieved. It was very difficult to completely remove fouling organisms, especially in areas next to ropes. As a consequence, fouling re-growth rate was fast in C2 and especially in C3, where the coating did not contain copper at all. Under these conditions, Percentage Net Occlusion was around 28% and 45%, respectively, just before every cleaning operation. In contrast, PNO in C1 at the end of the experiment was around 13%. Occlusion levels in C2 and C3 suggest that net cleaning should ideally have been done more often as stated before [3]. Nevertheless, a high cleaning frequency might have a direct effect on fish and net performance, which together with higher operation expenses, would increase fish production cost.

The industry uses coatings to protect net fibers from abrasion produced by high pressure water disks in the nets designed to be cleaned on-site. Moe et al. [19] concluded that antifouling treatments reduce the tensile strength of netting. However, in this study, after seven months in real working conditions, it seems like both antifouling and coating treatment protected the netting to some extent from external agents. This discrepancy could be explained by the fact that the direct effect of a coating or paint as such on polyamide fibers’ tensile strength can be detrimental for the direct effect of the paint/coating on the fibers; in actual working conditions, the layer coating on the netting could protect it from external agents such as UV radiation, abrasion, or temperature changes. When referring to protection against abrasion, the polyurethane coating used in C2 and C3 did not provide enough degree of protection since there was a dramatic loss of tensile strength in C2 and C3 nets after six episodes of in situ cleaning. This suggests that nets should be better protected or that an alternative to high pressure cleaning ought to be found. Currently, the use of low pressure, cavitation, and suction cleaning alternatives and high-performance coatings are being explored (unpublished data).

In terms of production performance, handling and farm operations generate stress and have a direct impact on fish [20,21]. On-site cleaning of the nets was originally thought to be a low invasive practice that would not cause stress in fish. In this trial, final weight achieved in C2 and C3 was significantly lower that growth in C1. Looking at the evolution of growth along the experiment, it seems that on-site cleaning operations during the experiment could effectively have exerted an effect on fish growth, and this would be in contraposition with the initial hypothesis. Considering that the cleaning rate should have probably been higher, its effects in terms of stress and performance would have been even more severe. Cleaning operations may also affect feeding patterns and behavior of fish. In this trial, fish were fed before net cleaning, but feeding response was very poor after the cleaning operation, even when the next meal was the following day. In fish farming, it might take some time to restore a regular feeding pattern after a disturbance. This can very easily result in a loss of fish growth. Compared to what would have been expected, growth attained was also slightly lower, even in C1. A factor that could have modulated the results is temperature. The fact that the trial was carried out in a shallow pond involved higher temperature variations than those at open sea. The negative effects of temperature variations on fish growth have already been described [22]. In fact, the feeding response under these high temperature fluctuation periods was very poor in all cages.

The results of this work suggest that on-site cleaning effect on gill integrity and health status are much more detrimental that those produced by net changing. When nets were cleaned in situ, fouling particles were washed off the net and a thick cloud of debris was generated, as previously described [23]. This organic matter remains in and around the cage for a certain period of time, depending on current direction and intensity, and eventually passes through the gills. This not only has an effect on gill integrity, but can also generate a stress response in fish due to turbidity generated, noise, or even the presence of the cleaning equipment itself. In this study, on-site cleaning caused an array of typical detrimental effects on gill health status, similar to effects previously described in other studies [24,25], (Figure 5).

It is important to point out that in this work, small cages were used and it could have potentiated the effects of on-site cleaning in fish. It should be borne in mind that the ratio between net surface and net volume reduces as the cage/net size increases, and so the effects of net cleaning could be lower in commercial cages than that obtained in this work.

A literature review suggests that fouling in aquaculture cages and nets may act as a reservoir for parasites, increasing disease risk [4,6,26,27,28,29,30]. Interestingly, gill parasite *Sparicotyle chrysophrii* was found in fish in the C2 and C3 groups, whereas none was found in C1 gills. As has been stated previously [30], gill parasite presence is unusual in fish reared in tanks or ponds, probably because of turbidity or temperature variations. This could explain the low presence of parasites detected in this study. However, some parasites were found, and the fact that their presence was only recorded in C2 and C3 could be related to the substantial degree of biofouling during the experiment, confirming thus the above-mentioned hypothesis. The same principle, together with the biocide effect of copper would explain why skin microbiota behaved differently in C1 than in C2 and C3 along the experiment. Biodiversity richness tended to decrease at the end of the experiment compared to the start in C1. On the other hand, in C2 and C3, microbiota richness was higher at the end than at the beginning of the experiment. Regardless that bacterial proliferation around the cage and in the skin could have positive or detrimental effects on fish depending on bacterial populations, it seems to be positively related with the presence of fouling on the nets.

The presence of parasites on the gills and, the higher degree of bacterial diversity due to on-site cleaning operations could be involved with the fact that liver and spleen organo-indexes were significantly higher in C2 and C3 compared to C1 at the end of the experiment. These organs are involved in detoxification and immune responses. Consequently, the conditions forementioned in C2 and C3 could promote a response in both tissues that could explain these differences. However, looking specifically at the liver, the results show a reduction in size in all three conditions, comparing the final with the initial weight. This could be associated to reserves mobilization during winter period, suggesting that resources from feed intake would not entirely cover the energetic demand. In this experiment, feeding was ad libitum 5 over 7 days per week, but as stated before, daily temperature variations affected appetite. However, in C1, growth achieved after seven months was only 6% lower than the expected rates in a commercial batch. Visceral fat levels were reduced compared to the reared animals. Since the final growth under our experimental conditions were close to what should be expected under commercial standards, the results suggest that overfeeding is generated during or before winter in sea bream fish farms that is not used for protein growth and is accumulated as visceral fat. This fact may be related to the winter syndrome affecting sea bream and showing similar symptoms [31].

Heavy metals bioaccumulation in fish tissues is a concern as it can affect consumer health. Several guidelines, regulatory frameworks, and government regulations exist to control the content of heavy metals in edible fish and aquaculture practices [32,33]. Commonly, copper content in fish is not a major issue since levels either in wild or reared fish are low and as other vertebrates, fish have very well developed copper homeostatic mechanisms. Several authors have assessed and compared copper bioaccumulation in wild and aquaculture fish. [34] reported even less copper content in muscle of farmed *Sparus aurata*, (1.3 ± 0.3 µg·g^−1^) than in wild caught specimens (1.6 ± 0.4 µg·g^−1^) in the Ligurian sea. In fish, waterborne copper can enter through the gills [11] but liver is the place where the metal can be found in higher concentrations due to the detoxification function of the tissue [35]. In this experiment, the use of cuprous oxide caused a significant degree of copper bioaccumulation after seven months. Especially in animals in C1, the levels in both liver and gills at the end of the experiment were significantly higher than those measured at the beginning. Despite the fact that C2 and C3 contained a small amount or none contain copper in their coatings, levels of copper in the liver at the end of the experiment were slightly higher than the levels at the beginning whereas no significant changes were detected in gills. In all cases, bioconcentration factor (BCF) was very low, always below 1 [36]. It is also important to remark that in all cases, copper amount recorded in tissues was well below food safety recommended levels [37]. Looking specifically at the content in muscle, in all three conditions, the values were lower than 0.25 µg·g^−1^, i.e., below the technique detection threshold. The significant increase of copper in liver in C2 and C3 could be explained by the fact that although cages layout followed the flow across the pond and there was a distance of three meters among one and another row of cages, copper contained in the antifouling paint in C1 may have reached the other cages. Additionally, the intimate relationship between the animals and the aquatic medium in which they are reared could have facilitated this contact.

## 5. Conclusions

According to our results, several issues need to be solved before considering on-site cleaning as a cheaper and less impactful alternative to the use of copper in aquaculture nets. Cleaning performance should improve to avoid a rapid fouling re-growth rate which nowadays, as here demonstrated, deems necessary a high cleaning frequency, involving an increase in operation costs and impact on fish health and performance. Nets should be better protected to withstand high water pressure abrasion or an alternative to high water pressure should be found. On the other hand, antifouling treatment offers better results both on fish and net performance. Besides, the use of copper should not be a concern for fish or consumer health since, after a rearing period of seven months, although some bioaccumulation occurred, levels of copper in tissues have been always found to be below safety thresholds for consumers, especially in muscles, where those levels were undetectable. The observed capacity of accumulating waterborne copper, BCF was extremely low, well below 1 in all tissues and specially in muscles. The intermediate alternative presented in this work (C2) is a promising option, since the results show the clear antifouling effect of the small amount of cuprous oxide contained in polyurethane coating, which would allow a lower frequency of cleaning operations. Overall, on-site cleaning technology as well as operational procedures must be improved to be considered as a future alternative.

## Figures and Tables

**Figure 1 animals-11-00734-f001:**
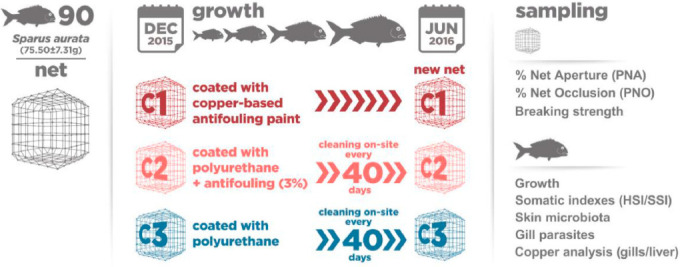
Experimental layout and system description.

**Figure 2 animals-11-00734-f002:**
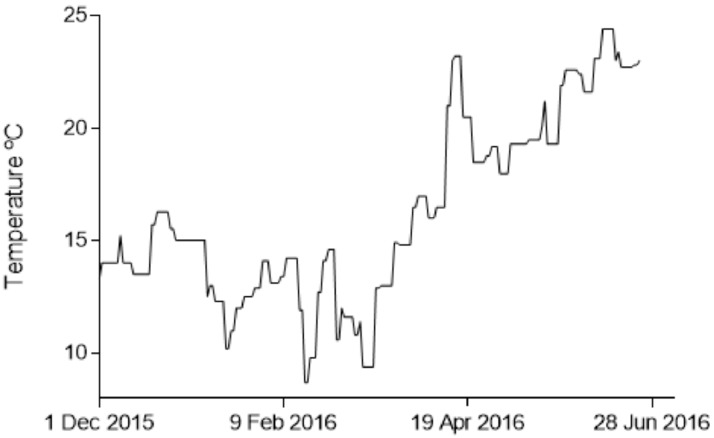
Registered pond temperature during the experiment.

**Figure 3 animals-11-00734-f003:**
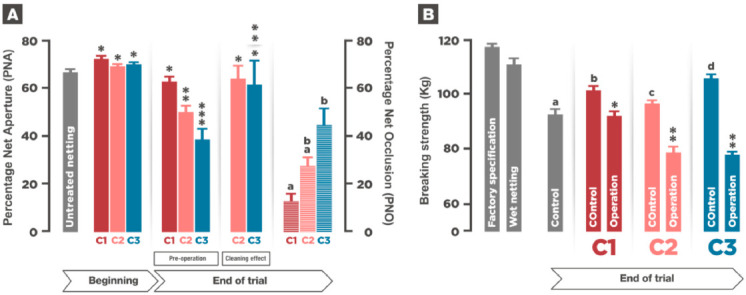
Net performance during the experiment. (**A**). Effect of netting treatment and PNA/PNO evolution during the experiment. Bars represent PNA and horizontal striped bars represent PNO (plotted on the right *Y* axis). At the beginning of the experiment (*n* = 8), there were no significant differences in PNA between treatments. Significant differences in PNA and PNO were recorded at the end of the experiment (*n* = 6) before cleaning operations between C1, C2, and C3 and within C2 and C3. Cleaning effect in C2 and C3 promoted PNA recovery to similar values at the end of the experiment. Values are means of three replicates. (**B**) Results of the BS tests performed during the experiment. Bars represent BS. Grey bar at the end of the experiment represents the BS of the untreated netting. During the experimental time, treated netting displayed significantly better resistance than the untreated netting (control). Cleaning effect along the experiment produced a loss of tensile strength in all three conditions. Values are means of three replicates, in Control columns (Control, Co) *n* = 45, in Operation columns (Op) *n* = 30. In all cases, error bars represent SE. Different numbers, letters and asterisks denote significant differences (*P* < 0.05) obtained by One-way ANOVA and a posterior Tukey’s test.

**Figure 4 animals-11-00734-f004:**
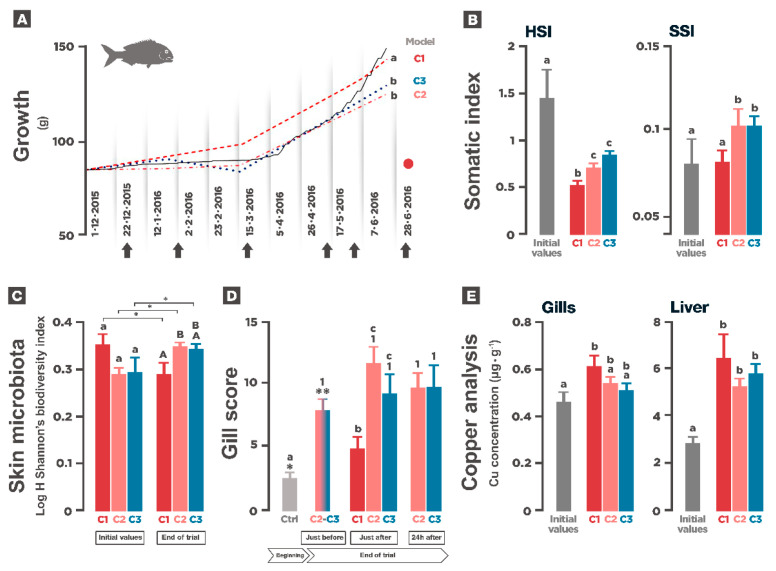
Fish performance during the experiment. (**A**). Growth evolution during the experiment. The line represents theoretical growth according to the growth model. The stapled line represents C1, stapled-dotted line C2 and dotted line C3. Arrows are on-site cleaning in C2, C3 and dot stands for net changing in C1. Values are means of three replicates. (**B**). Graphs show HSI and SSI at the beginning and at the end of the experiment. Initial and final values of hepatosomatic (HSI) and spleen-somatic (SSI) indexes. Initial values come from a pooled population of animals from the experimental groups. Final values are means of three replicates (*n* = 27). In b error bars represent SE. In a and b, letters denote significant differences (*P* < 0.05) obtained by One-way ANOVA and a posterior Tukey’s test. (**C**). Skin microbiota richness at the beginning and at the end of the experiment. Expressed as log H (Shannon’s index H). Values are means of three replicates *n* = 18. Error bars represent SE. Shared lowercase letters at the beginning and upper case letters at the end of the experiment denote insignificant differences. Asterisks denote significant differences within treatments between initial and final values, *P* < 0.05 obtained by GLM and one-way ANOVA and a posterior Tukey’s test. (**D**). Gill score at the end of the experiment. Effect of on-site cleaning in gill disorders. The grey bar represents the control. The pink-blue bar represents the mean value for C2 and C3 at the end of the experiment (after five cleaning episodes and before the last one). Error bars are SE. The asterisks explain differences between control values and gill score before net cleaning in C2 and C3. The letters explain differences between the control values and those after cage operation at the end of the experiment, and the numbers explain differences between C2 and C3 just after and 24 h after net cleaning at the end of the experiment. *P* < 0.05 obtained by GLM and One-way ANOVA and a posterior Tukey’s test. (**E**). Copper content in liver and gills at the beginning and at the end of the experiment. Livers values are plotted on the left *Y* axis, while gill values are plotted on the right *Y* axis. Significant differences are shown in lower capital letters in liver and in capital letters in gill. Different letters, numbers, and asterisks denote significant differences *P* < 0.05 obtained by GLM and One-way ANOVA and a posterior Tukey’s test.

**Figure 5 animals-11-00734-f005:**
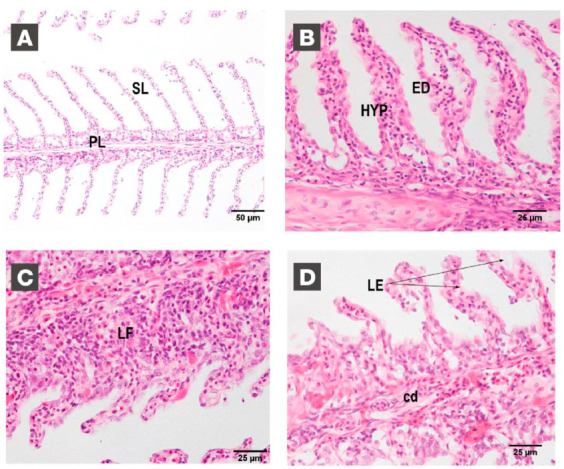
Gill damage at the end of the experiment. H-E stained sections of healthy and damaged gills. (**A**), non-damaged gill at the beginning of the experiment with well-structured primary lamellae (PL) and secondary lamellae (200×) (SL). (**B**–**D**) Different examples of gill disorders at the end of the experiment (400×). Secondary lamellae hyperplasia (HYP) and edema (ED), lamellar fusion (LF), and loss of structure (LE) as well as severe circulatory disruptions (cd) are labelled in the figures.

**Table 1 animals-11-00734-t001:** Water parameters monitored in the pond during the experiment.

Temperature °C	O_2_ ppm	O_2_ SAT. %	NO_2_− mg·L^−1^		NH_4_+ mg·L^−1^	pH	Salinity ‰
Average	16.61 ± 3.95	11.99 ± 4.31	124.06 ± 40.05	0.19 ± 0.18	0.05 ± 0.07	8.32 ± 0.19	28.99 ± 4.11
Min.	*8.7*	*5.2*	*70.0*	*0.00*	*0.00*	*8.00*	*14.40*
Max.	*24.4*	*24.2*	*249.0*	*0.80*	*0.70*	*8.80*	*34.80*

Values are means ± SD. Rows in italics show lowest and highest peaks recorded during the experiment.

**Table 2 animals-11-00734-t002:** *Sparicotyle chrysophrii* presence in gills at the end of the trial.

*P* (%)	A		MI
Condition 1	0.00	0.00	0.00
Condition 2	25.00	0.83	3.30
Condition 3	40.00	0.80	2.00

*P*. Prevalence, A. Abundance, MI. Mean Intensity.

## Data Availability

Data generated in this study is available at DDD repository (Digital Deposit of Documents, Universitat Autonoma de Barcelona, Spain).

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
