# Peer review of "Effects of Fouling Management and Net Coating Strategies on Reared Gilthead Sea Bream Juveniles"

_animals, 2021, doi:10.3390/ani11030734_

Round 1

Reviewer 1 Report

The article is interesting, worth to publish after the proposed corrections are made. I have indicated comments and suggestions in the text of the paper.

Author Response

Response to reviewer 1

Notation mistakes have been corrected as well as other comments within the text. Bioconcentration factor of copper has been calculated and cited in the text.

Regarding sample size when assessing copper bioaccumulation, we agree with the reviewer that 3 animals per condition would be insufficient. However we triplicate the cages, so each condition finally included 9 animals. This sort of arrangement has been also followed by other authors. Regarding the Cu content of muscle, the apparatus gave values below the detection threshold, so we are sorry, but we have not this graph anymore.

Reviewer 2 Report

The manuscript “Effects of fouling management and net coating strategies on reared gilthead sea bream juveniles” authored by J. Comas, D. Parra, J.C. Balasch and L. Tort  presents a study in which the authors compared the polyurethane coating and the use of cuprous oxide as antifouling strategies for nets management in aquaculture. The topic is relevant, and the scientific approach is rigorous. The obtained results have been well presented and discussed.

Only minor revisions should be made as follows:

Simple summary

The manuscript misses the simple summary.

Introduction

Line 30

“It pretends”. Please use another verb.

Lines 106, 112 and throughout the text

Check and in case write properly the chemical formula and the units (Cu2O, m2….etc.)

Results

Table 1

Please check the values of O2 Sat % and NO2 mg/L. Only the 24% of O2 maximum value? Possible? And for nitrite the values seem to be much higher than the acceptable limits for the fish survival (usually < 0.1 mg/L). In the same table use periods and not commas to indicate decimals.

Citations and references throughout the manuscript should be adapted to the journal standards following the authors guidelines.

Author Response

Response to reviewer 2.

Simple summary

The manuscript misses the simple summary.

Simple summary has been added

Introduction

Line 30

“It pretends”. Please use another verb.

Sentence has beer re-writen

Lines 106, 112 and throughout the text

Check and in case write properly the chemical formula and the units (Cu2O, m2….etc.)

Notation amended

Results

Table 1

Please check the values of O2 Sat % and NO2 mg/L. Only the 24% of O2 maximum value? Possible? And for nitrite the values seem to be much higher than the acceptable limits for the fish survival (usually < 0.1 mg/L). In the same table use periods and not commas to indicate decimals.

Max O2 sat in the table is 124%

O2 ppm max value is 24,2

0,8 mg/L-1 was a very punctual record and this value does not pose in risk sea bream as it is a specially tolerant species. See:

Acute Toxicity of Ammonia and Nitrite to Sea Bream, Sparus aurata (Linnaeus, 1758), in Relation to Salinity

  • August 2017
  • Journal of the World Aquaculture Society49(1)

DOI: 10.1111/jwas.12448